# How Should Corruption Be Used in SSL? Empirical Insights for Effective Pretraining

## Abstract

We study how corruption design—masking and additive noise—affects self-supervised pretraining of vision models. Although denoising diffusion models succeed in generation, noise-driven extensions of masked image modeling (MIM) achieve only marginal gains on recognition tasks, including fine-grained benchmarks. We thus investigate why this would be the case, seeking effective ways to combine masking and noising within the corruption-to-reconstruction (C2R) paradigm. We begin by analyzing prior noise-based MIM approaches, categorizing them into Substitutive Corruption (masked tokens replaced by noised ones) and Conjunctive Corruption (masked and noised tokens coexist), and further into Encoder- or Decoder-style depending on where corruption and restoration occur. Our study reveals that the literature trends toward a Decoder-style design. In contrast, we evaluate an Encoder-style alternative with a focus on transfer. Building on these analyses, we propose three principles for effective C2R pretraining: corruption and restoration should occur within the encoder, noise is most effective when injected at the feature level, and mask reconstruction and de-noising must be explicitly disentangled to avoid interference. By implementing these findings, we propose a framework that captures a broader frequency spectrum of representations and improves transferability, surpassing MIM by up to 8.1% and recent noise-driven pretraining methods by 8.0% across diverse recognition benchmarks. **Code** is available in the Supplementary Material.

## 1 Introduction

Self-supervised learning (SSL) has emerged as a key paradigm in computer vision, enabling the pretraining of large-scale models (Dosovitskiy et al., 2020; Liu et al., 2021) on massive unlabeled datasets and transferring them to diverse downstream tasks (Chen et al., 2020; He et al., 2020; Grill et al., 2020; Bao et al., 2021; He et al., 2022; Xie et al., 2022). By removing the reliance on costly annotations, SSL has alleviated the data-hungry nature of foundational models and driven progress in image classification, semantic segmentation, object detection, and fine-grained recognition (Carion et al., 2020; Ranftl et al., 2021; Zhu et al., 2020; Zheng et al., 2021; Chen et al., 2021), establishing itself as a cornerstone of representation learning.

A dominant line of SSL research follows the **corruption-to-reconstruction (C2R)** paradigm, where inputs are intentionally corrupted and the model is trained to reconstruct the original data. Masked image modeling (MIM) exemplifies this strategy (Bao et al., 2021; He et al., 2022; Xie et al., 2022), masking large portions of input patches to encourage spatial reasoning and semantic understanding. These methods have shown remarkable effectiveness and scalability, achieving state-of-the-art results on various vision benchmarks (Deng et al., 2009; Zhou et al., 2017; Lin et al., 2014).

Meanwhile, motivated by the success of generative models such as denoising diffusion models (Ho et al., 2020; Rombach et al., 2022; Ramesh et al., 2021; Saharia et al., 2022), recent works have explored noise-based C2R pretraining. DiffMAE (Wei et al., 2023) and MaskDiT (Zheng et al., 2023) extend masking-based pretraining by introducing noise in different ways. DiffMAE (Wei et al., 2023) replaces masked tokens with noised ones, and MaskDiT (Zheng et al., 2023) further leverages both; collectively, they illustrate the integration of masking and noising within a unified pretraining framework. While diffusion models excel in high-fidelity image generation (Dhariwal & Nichol, 2021), these noise-driven approaches (Wei et al., 2023; Zheng et al., 2023) do not provide

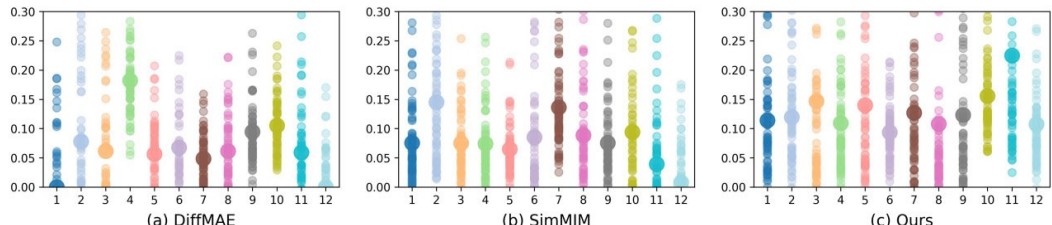

Figure 1: We plot the KL divergence of attention distributions across heads (small dots) and their layer-wise means (large dots) for (a) a recent noise-based MIM method (Wei et al., 2023), (b) a representative MIM method (Xie et al., 2022), and (c) ours. Higher KL divergence indicates broader frequency coverage. Our method achieves greater diversity than MIM and noise-based baselines, accounting for its strong performance on recognition tasks, including fine-grained settings.

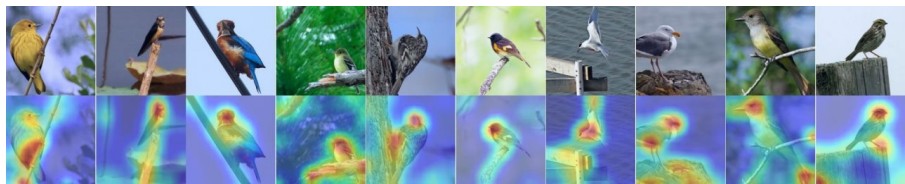

Figure 2: We visualized the self-attention maps for the image classification token in the final layer of our model on a fine-grained visual categorization benchmark. The proposed method captures a range of frequencies by focusing on both key features and fine details within complex scenes.

notable gains over MIM on *recognition benchmarks* (Wah et al., 2011; Van Horn et al., 2015; 2017; 2018; Krause et al., 2013; Maji et al., 2013; Deng et al., 2009; Zhou et al., 2017; Lin et al., 2014), spanning both fine-grained recognition and general vision tasks. This indicates that although noise introduces high-frequency variations, such information is not effectively encoded into transferable representations for recognition tasks.

In Sec. 3, we systematically analyze why noise-based C2R pretraining yields limited gains. We first dissect their design choices, categorizing them into two paradigms: *Substitutive Corruption*, which replaces masked tokens with noised ones, and *Conjunctive Corruption*, where masked and noised tokens coexist. We then classify these paradigms into *Encoder-* and *Decoder-styles* depending on where corruption and reconstruction occur, noting that prior methods are largely Decoder-style. Building on these analyses, we propose three design principles for effectively unifying masking and noising: (1) **corruption and restoration should occur within the encoder**, as the encoder is what is ultimately transferred to downstream tasks; (2) **noise is most effective when injected at the feature level**, particularly in lower encoder layers, where high-frequency details are present; and (3) **masked token reconstruction and de-noising must be explicitly disentangled to avoid interference**, which we enforce by suppressing attention between the two token types.

With these findings, we design a novel pretraining setup that effectively utilizes both masking and noising. Our approach captures a richer frequency spectrum of image representations as shown in Figs. 1 and 2, enhancing transferability across a variety of downstream tasks and recognition benchmarks; CUB-200-2011 (Wah et al., 2011), NABirds (Van Horn et al., 2015), iNaturalist 2017/2018 (Van Horn et al., 2017; 2018), Stanford Cars (Krause et al., 2013), Aircraft (Maji et al., 2013), ImageNet (Deng et al., 2009), ADE20K (Zhou et al., 2017), and COCO (Lin et al., 2014). Our method achieves up to 8.1% performance gain over MIM baselines and 8.0% over recent noise-driven pretraining methods, validating the effectiveness of our design.

To summarize, our contributions are.

- We provide a thorough empirical study on why current noising-based pretraining approaches (Wei et al., 2023; Zheng et al., 2023) do not provide noticeable gains for recognition tasks.

- We provide guidelines from our detailed study on how to use corruptions within pretraining.

- With our findings, we propose a novel pretraining method that outperforms the state-of-the-art on a wide range of recognition tasks, including fine-grained tasks.

## 2 PRELIMINARY AND RELATED WORKS

Since the intuitions and findings of our work build upon masked image modeling (MIM) and denoising diffusion models, we first revisit these foundations for completeness.

### 2.1 MASKED IMAGE MODELING (MIM)

The core idea of MIM is to randomly mask a subset of image tokens and train the model to reconstruct the missing content in a self-supervised manner.

**Random masking.** Formally, let $X \in \mathbb{R}^{N \times L \times D}$ denote the input sequence of image tokens, where $N$ is the batch size, $L$ the number of tokens per image, and $D$ the token dimension. We define the mask generation process as $M = \Phi_M(X, \gamma)$, where $\gamma$ is the masking ratio and $M \in \{0, 1\}^{N \times L}$ indicates a mask map. The masking operation generates a corrupted signal $X_{cor}$ as

$$X_{cor} = X \odot M + \theta \odot (1 - M), \tag{1}$$

where $\odot$ denotes the Hadamard product, and $\theta$ is a learnable parameter for masked tokens.

**Reconstruction.** To learn to reconstruct the original tokens $X$ back from only the visible ones $X_{\text{vis}}$, training of MIMs typically relies on mean squared error (MSE). With the token predictions $\hat{X}$ from MIM framework that takes as input the masked visible tokens $X_{\text{masked}}$, we minimize the loss:

$$\mathcal{L}_{\text{MIM}} = \frac{1}{\sum_{k,l}(1 - M_{k,l})} \sum_{k=1}^{N} \sum_{l=1}^{L} (1 - M_{k,l}) \|\hat{X}_{k,l} - X_{k,l}\|^2. \tag{2}$$

**Recent works.** MIM approaches (He et al., 2022; Xie et al., 2022; Choi et al., 2024; 2025; Bao et al., 2021; Yi et al., 2022; Dong et al., 2022; Chen et al., 2024a; Assran et al., 2023) adapt the concept of Masked Language Modeling (MLM) from NLP. BEiT (Bao et al., 2021) applies MLM-like pretraining to images using discrete visual tokens generated by a pre-trained dVAE. MAE (He et al., 2022) focuses only on visible patches in the encoder, predicting masked pixel values through a decoder. SimMIM (Xie et al., 2022) uses both visible and masked patches in the encoder and predicts original pixels directly. Recent advances (Choi et al., 2024; 2025) focus on masked tokens for fast convergence and performance improvement.

### 2.2 DENOISING DIFFUSION MODEL

Denoising diffusion models are trained by progressively corrupting inputs with Gaussian noise and learning to denoise them. This is in a similar spirit to MIMs, but unlike MIMs, the theoretical foundations allow the model to generate new data, hence they are generative (Song et al., 2020).

**Forward diffusion.** Forward diffusion iteratively adds noise to an input image sequence $X \in \mathbb{R}^{N \times L \times D}$ over $T$ time steps. At each time step $t$, a noise schedule $\beta^t \in \mathbb{R}$ controls the amount of noise added, where $\beta^t$ is a scalar that determines the noise level at time step $t$. The corrupted representation $X^t$ at step $t$ is then defined as:

$$X^t = \sqrt{1 - \beta^t} \cdot X^{t-1} + \sqrt{\beta^t} \cdot \epsilon, \tag{3}$$

where $\epsilon \sim \mathcal{N}(\mathbf{0}, \mathbf{I})$ is the Gaussian noise with a zero matrix $\mathbf{0}$ and an identity covariance matrix $\mathbf{I}$. This iterative process gradually *diffuses* the data towards Gaussian noise as $t$ approaches $T$.

**Denoising.** With the corrupted signal, the denoiser then learns to undo this corruption, effectively allowing the model to traverse back through the diffusion process, that is, transform Gaussian noise to clean data that follows the data distribution. Specifically, starting from $X^T$, the model learns to predict the clean image $X^0$ by estimating the intermediate states through a denoising function

$\Phi_{\text{denoise}}$, which is typically parameterized as a neural network. The denoising step at time $t$ can be represented as:

$$\hat{X}^{t-1} = \Phi_{\text{denoise}}(X^t, t), \tag{4}$$

where $\hat{X}^{t-1}$ represents the denoised estimate at time $t - 1$. Training of $\Phi_{\text{denoise}}(X^t, t)$ is performed through various variations of the original DDIM (Song et al., 2020) and DDPM (Ho et al., 2020) methods, including recent family of Rectified Flow models (Liu et al., 2022), but these approaches are all essentially focusing on obtaining $\hat{X}^{t-1}$ estimates in some form that will accurately lead toward $X^0$ through various solvers (Lu et al., 2022; Karras et al., 2022).

**Recent works.** Denoising diffusion models (Ho et al., 2020; Nichol & Dhariwal, 2021; Rombach et al., 2022; Ramesh et al., 2021; Saharia et al., 2022) have gained prominence in generative tasks for their ability to produce high-quality, detailed images. DDPM (Ho et al., 2020) introduced the foundational framework, where Gaussian noise is gradually added to an image and then removed in a reverse process, Improved DDPM (Nichol & Dhariwal, 2021) enhanced this approach with modifications in noise scheduling and model architecture. LDM (Rombach et al., 2022) further improved efficiency by operating in a compressed latent space rather than pixel space, allowing various practical applications.

## 2.3 PRETRAINING VIA DENOISING

With preliminaries on MIMs and denoising diffusion models, we now review two representative works that aim to marry the two schools of thought into a single pretraining framework.

**DiffMAE.** DiffMAE (Wei et al., 2023) combines diffusion-based modeling with MIM. Instead of replacing masked patches with a learnable token, DiffMAE corrupts them by progressively injecting Gaussian noise following a predefined schedule $\alpha_t \in \mathbb{R}$. Given a binary mask $M$, Gaussian noise $\epsilon \sim \mathcal{N}(\mathbf{0}, \mathbf{I})$ is progressively added to the selected tokens over $T$ steps, formulated as:

$$X_{cor} = X_v + X_n^t = X \odot M + \left(\sqrt{\alpha_t} \cdot X + \sqrt{1 - \alpha_t} \cdot \epsilon\right) \odot (1 - M). \tag{5}$$

For clarity, we denote the *visible tokens* as $X_v$ and the *noised tokens* as $X_n^t$. The model then reconstructs $X$ by denoising the corrupted tokens $X_n$ through an iterative reverse process conditioned on the visible tokens $X_v$:

$$\hat{X}_n^{t-1} = \Phi_{\text{denoise}}(X_n^t, X_v, t), \tag{6}$$

where $\Phi_{\text{denoise}}$ denotes the denoising function.

**MaskDiT.** MaskDiT (Zheng et al., 2023) introduces a noise-based pretraining approach that leverages masked transformers for faster training of diffusion models. Unlike DiffMAE, MaskDiT generates a corrupted input using both *noised tokens* and *masked tokens*:

$$X_{cor} = X_n^t + X_m = \left(\sqrt{\alpha_t} \cdot X + \sqrt{1 - \alpha_t} \cdot \epsilon\right) \odot M + \theta \odot (1 - M), \tag{7}$$

where $\theta$ denotes a learnable parameter for the masked tokens. Similarly, as before, we denote the *noised tokens* $X_n^t$ and the *masked tokens* $X_m$. The model then learns to reconstruct both the noised tokens $X_n^t$ and the masked tokens $X_m$ via denoising and reconstruction function $\Phi$:

$$(\hat{X}_n, \hat{X}_m) = \Phi(X_n^t, X_m, t). \tag{8}$$

## 3 AN ANALYSIS OF PRIOR PRETRAINING METHODS

Recent noise-based C2R methods (Wei et al., 2023; Zheng et al., 2023) augment masked image modeling (MIM) by injecting additive Gaussian noise to capture fine-grained detail. Yet, consistent with the results reported in the prior study (Zheng et al., 2023), our experiments show no notable gains over MIM baselines (Xie et al., 2022; He et al., 2022) on recognition tasks (Fig. **??**). Under matched pretraining and identical fine-tuning, both variants perform on par with MIM on ImageNet (Deng et al., 2009) and underperform on FGVC (Wah et al., 2011; Van Horn et al., 2015; 2017; 2018; Krause et al., 2013; Maji et al., 2013), where fine detail matters most. Simply adding a denoising stage to MIM does not improve representation quality for recognition. We therefore examine *how* masking and noising are combined and *where* corruption is applied.

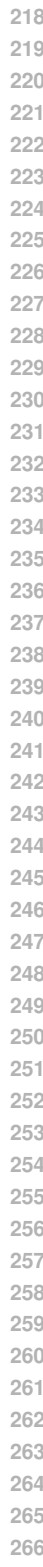
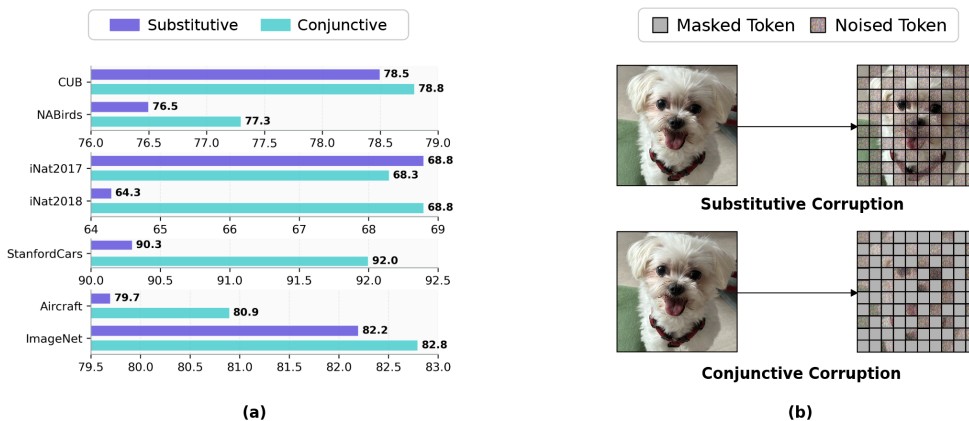

Figure 3: **(a)** Conjunctive Corruption achieved better performance across datasets, as it consistently retains a fully masked portion that enhances semantic discriminability. In contrast, Substitutive Corruption relies on tokens with random noise intensities, which may limit its effectiveness. **(b)** Illustration of two corruption paradigms: Substitutive Corruption, where masked tokens are replaced by noised ones; and Conjunctive Corruption, where masked and noised tokens coexist.

### 3.1 How should masking and noising be combined?

We first examine how recent methods integrate noising with masking to investigate why they provide limited gains on recognition tasks. Specifically, we look into the two representative cases of integrations, DiffMAE (Wei et al., 2023) and MaskDiT (Zheng et al., 2023).

- **Substitutive Corruption**: masked tokens are *replaced* with noised tokens;
- **Conjunctive Corruption**: masked tokens are *retained* while visible tokens are additionally noised.

*Substitutive Corruption* (Wei et al., 2023) employs a noised token alongside a clean visible token, as specified in (5), and the model focuses on denoising (6). On the other hand, *Conjunctive Corruption* (Zheng et al., 2023) utilizes both a masked token and a noised token, as described in (7), and the model performs both denoising and reconstruction (8). Fig. 3 (b) illustrates these two alternatives.

We evaluated the two corruption methods used in recent baselines (Wei et al., 2023; Zheng et al., 2023). Fig. 3 (a) presents the transfer learning performance measured after pretraining on ImageNet-1K (Deng et al., 2009) and fine-tuning across recognition benchmarks (Wah et al., 2011; Van Horn et al., 2015; 2017; 2018; Krause et al., 2013; Maji et al., 2013), confirming that Conjunctive Corruption consistently outperforms Substitutive Corruption. We attribute this to the limitation of Substitutive Corruption, which relies solely on noise as a corruption. Since random time sampling often produces nearly clean inputs, the pretraining task may become trivial and fail to encourage meaningful semantic learning. In contrast, Conjunctive Corruption always retains masked regions, compelling the model to jointly solve denoising and reconstruction, encouraging richer feature representations.

### 3.2 Where should corruption be applied?

Beyond the strategy to combine masking and noising, a further key question is *where* corruption should be applied. **We first categorize MIM paradigms into two types** based on the placement of masked tokens, as illustrated in Fig. 4 (a):

- **Encoder-style** (Xie et al., 2022; Bao et al., 2021; Yi et al., 2022): masked tokens are injected *into the encoder*, and reconstruction is performed across the encoder–decoder.
- **Decoder-style** (He et al., 2022; Chen et al., 2024a; Dong et al., 2022): masked tokens are processed only *in the decoder*, while the encoder learns solely from visible tokens.

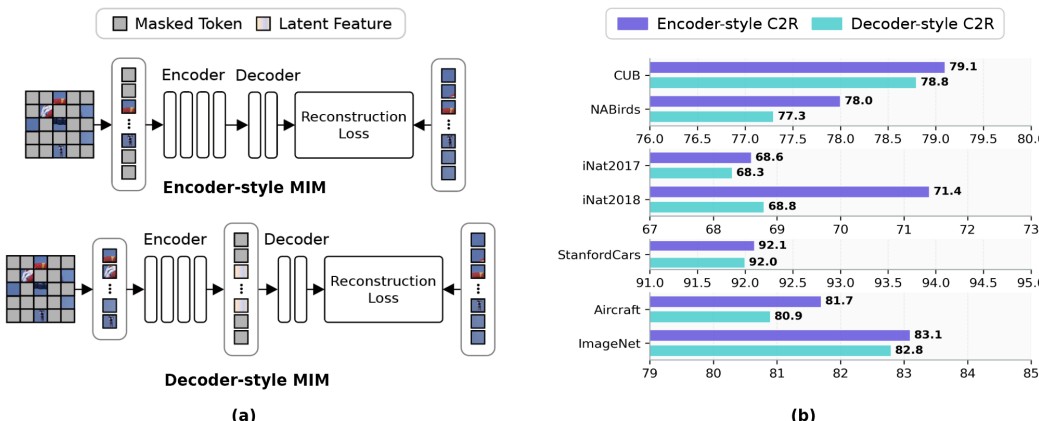

Figure 4: **(a)** The study of MIM is broadly segmented into two types based on masked token placement: Encoder-style, which reconstructs masked regions within the encoder (Xie et al., 2022; Bao et al., 2021; Yi et al., 2022; Choi et al., 2024), and Decoder-style, where reconstruction occurs solely in the decoder (He et al., 2022; Chen et al., 2024a; Dong et al., 2022). The recent noise-based C2R baselines (Wei et al., 2023; Zheng et al., 2023) build on MAE (He et al., 2022), can be seen as Decoder-style. **(b)** We implemented naive Encoder- and Decoder-style C2R frameworks, both with Conjunctive Corruption. Consistent with our hypothesis, the Encoder-style performs better across standard and fine-grained tasks, though the gains are modest, suggesting that naive implementations fall short. We thus further examine how Encoder-style design can more fully exploit its advantages in Sec. 4.1.

Recent noise-based pretraining approaches (Wei et al., 2023; Zheng et al., 2023) primarily adopt a Decoder-style design built on MAE (He et al., 2022).

However, it is important to note that only the encoder is transferred for downstream fine-tuning. This suggests that the placement of corruption and reconstruction may matter and has motivated MIM baselines to explore Encoder-style designs; accordingly, recent works (Xie et al., 2022; Bao et al., 2021; Yi et al., 2022; Choi et al., 2024) adopts an Encoder-style design. We therefore study an Encoder-style variant that applies corruption and reconstruction inside the encoder, such that noising directly shapes the learned transferable representations. The next section (Sec. 4.1) details this design and compares it head-to-head with matched Decoder-style baselines.

# 4 PROPOSED METHOD

Building on the analysis of prior methods, which revealed strengths and limitations of existing designs, we move beyond them and identify three novel design principles to effectively unify masking and noising. These principles, detailed in the following subsections, are as follows:

- **Encoder-style**: corruption and restoration should occur within the encoder;
- **Feature-level noise**: noise is most effective when injected at the feature level; and
- **Task disentanglement**: masked token reconstruction and de-noising must be explicitly disentangled.

## 4.1 CORRUPTION AND RESTORATION SHOULD OCCUR WITHIN THE ENCODER

In most transfer learning pipelines, the encoder is transferred and fine-tuned for downstream tasks, while the decoder is typically discarded. Consequently, when corruption is applied only at the latent level and restoration is confined to the decoder, the encoder neither learns to handle corrupted signals nor explicitly engages in restoration, limiting the relevance of its learned features. In contrast, introducing corruption and enforcing restoration within the encoder directly couples representation learning with corruption handling, which, in principle, should promote richer and more transferable

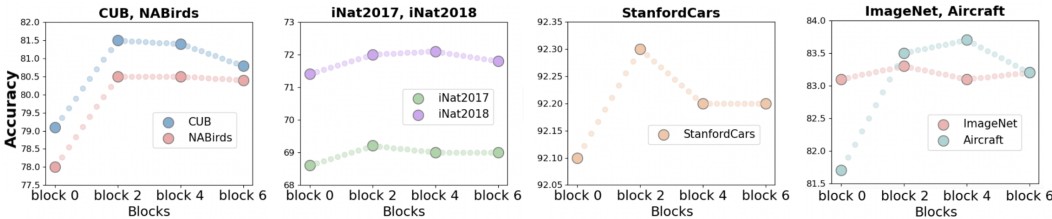

Figure 5: We introduced noise at encoder blocks 0, 2, 4, and 6. Feature-space injection (blocks 2, 4, and 6) outperformed pixel-space (block 0) on recognition tasks, with optimal performance at **block 2**, where high-frequency details are captured.

features. This theoretical motivation forms the basis of our hypothesis that Encoder-style corruption design offers clear advantages for pretraining. We then evaluated their performance on transfer learning by pretraining both models on ImageNet-1K (Deng et al., 2009) and fine-tuning them on the range of recognition tasks and datasets.

Fig. 4 (b) reports the transfer learning performance of each design. We implemented two naive noise-based frameworks featuring Conjunctive Corruption strategies that differ only in their placement of corruption. We kept all other factors identical. Consistent with our hypothesis, the Encoder-style structure performs better than the Decoder-style design across both standard recognition tasks and fine-grained benchmarks. However, the margin of improvement was smaller than anticipated, suggesting that a *naive implementation alone does not fully reveal its potential*. In the following subsection, we delve deeper into why this is the case and outline how the Encoder-style paradigm can better realize its advantages.

## 4.2 NOISE IS MOST EFFECTIVE WHEN INJECTED AT THE FEATURE LEVEL

Much of the success of denoising diffusion models is rooted in the application of noise at the latent (feature) space (Rombach et al., 2022; Chen et al., 2024b). While pixel-space diffusion exists (Hoogeboom et al., 2024), they need careful strategies on how noise should be applied. As such, we also suspect this to be the case for pretraining. However, in the naive implementation that we consider in Sec. 4.1, the Decoder-style method adds noise at the latent space immediately before the decoder, whereas the Encoder-style approach injects noise in pixel-space, prior to the encoder. We thus evaluate the Encoder-style setting by adding noise to various blocks within the encoder to investigate the impact of different noise-addition locations.

In Fig. 5, we conducted experiments by varying the stage at which noise is introduced within the encoder, specifically at different encoder blocks (blocks 0, 2, 4, and 6). The transfer learning performance results for recognition tasks verify that adding noise in feature-space (blocks 2, 4, and 6) is more effective than in pixel-space (block 0). Additionally, the highest performance observed at 'encoder block 2' suggests that noise addition is particularly effective when applied in the lower layers of the encoder, where high-frequency details are captured. This result reveals that the injecting noise at the feature level is crucial for maximizing the transfer learning potential of the model.

## 4.3 MASKED TOKEN RECONSTRUCTION AND DE-NOISING SHOULD BE EXPLICITLY DISENTANGLED

Referring to recent studies of MIM (Choi et al., 2024; He et al., 2022; Dong et al., 2022), Encoder-style approaches have shown mixed outcomes compared to Decoder-style. A plausible contributor is that masked tokens are optimized along directions weakly aligned with those of visible tokens (Choi et al., 2024), which can interfere with updates to visible token representations. Since Conjunctive Corruption uses masked tokens alongside noisy visible tokens, we hypothesize a similar risk in noise-based C2R pretraining, where masked tokens may interact undesirably with the encoding of noisy visible tokens.

To address this, we propose an explicit objective that disentangles the masked token reconstruction from the de-noising strategy. We introduce disruption loss, a variant of masked token optimization

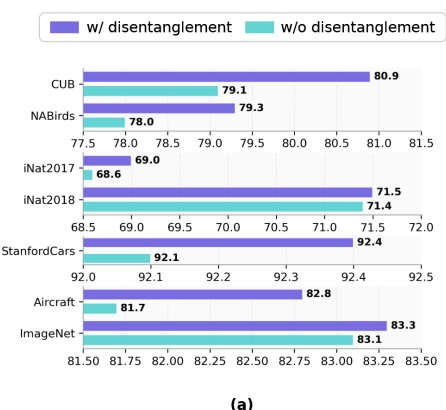 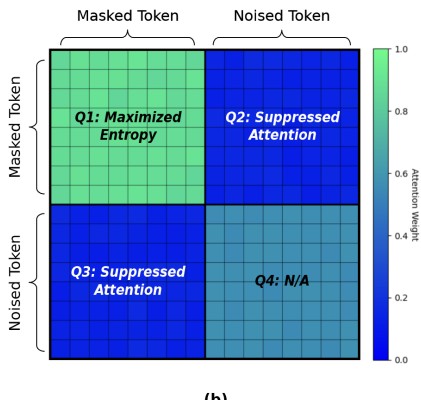

(a)  (b)

Figure 6: **(a)** We propose an explicit objective to disentangle the de-masking task from the de-noising task, fully harnessing both reconstructions within the encoder. The disruption loss adjusts the weight distribution of the affinity map, minimizing the influence of masked tokens on noisy visible tokens, and enhancing performance across both fine-grained and standard recognition tasks. **(b)** Disruption loss adjusts the affinity matrix to suppress masked-noised interactions, enforcing disentanglement between de-noising and mask reconstruction within the encoder.

proposed in MTO (Choi et al., 2024), designed to suppress attention between the two different token types. Disruption loss leverages per-row sparsities within the affinity matrix, which consists of four quadrants in Fig. 6 (b) as follows:

$$A = \begin{bmatrix} A_{mm} & A_{mn} \\ A_{nm} & A_{nn} \end{bmatrix}, \tag{9}$$

where $A_{mm}$ represents weights between masked-masked tokens, $A_{mn}$ and $A_{mn}$ represents affinities between masked-noised tokens, and $A_{nn}$ represents weights among noised tokens. The disruption loss $\mathcal{L}_d$ *suppresses the attention of the masked-noised tokens* by increasing the entropy of the attention between masked-masked token in the row unit of the affinity matrix. Thus, $\mathcal{L}_d$ recalibrates the weight distribution of $A$, minimizing the impact of masked tokens $x_m$ on noisy visible tokens $x_n^t$:

$$\mathcal{L}_d = -\sum_{i \in \mathcal{N}} \sum_j \tilde{p}_{i,j} \log \tilde{p}_{i,j} \tag{10}$$

where $\mathcal{N}$ denotes the index set of masked tokens $x_n^t$, and $\tilde{p}$ is the row-wise softmax of the affinity entries in $A$, satisfying $0 < \tilde{p}_{i,j} < 1$ and $\sum_j \tilde{p}_{i,j} = 1$. The application of $\mathcal{L}_d$ reduces interference between different token types and thus ensures effective task disentanglement between masked token reconstruction and denoising within the encoder.

The experimental results in Fig. 6 (a) exhibit a performance improvement from this weight adjustment on both fine-grained and standard recognition tasks. This demonstrates that explicitly separating de-masking and de-noising objectives maximizes the transferability of the Encoder-style approach.

## 5 EXPERIMENTS

### 5.1 IMPLEMENTATION DETAILS

All experiments in this manuscript were conducted under precisely identical conditions to ensure accurate analysis. For consistency, we evaluated all methods using official implementations (except DiffMAE, which we reimplemented due to lack of release) under our hardware configuration (4 × A100 GPUs). Since baselines have been trained in large cluster resources that are not available to everyone, reproduced results may differ from original papers. Please note that we ensured all comparisons followed the same setup, with code provided for verification in the Supplementary Material.

Table 1: To ensure statistical significance, we conducted 5 trials with different random seeds, reporting the mean (bars) and standard deviation (lines) for each method. The proposed method (Ours) consistently outperforms representative MIM (Xie et al., 2022; He et al., 2022) and noise-based methods (Wei et al., 2023; Zheng et al., 2023) across a wide range of recognition tasks, capturing diverse frequency details as shown in Fig. 1 and Fig. 2, that improve accuracy in FGVC, image classification, semantic segmentation, object detection, and instance segmentation tasks.

| Method | Fine-Grained Visual Categorization (FGVC) (Top-1 Acc %) | | | | | | Image Classification | Semantic Segmentation | Object Detection & Instance Segmentation | |
| | CUB | NABirds | iNat2017 | iNat2018 | StanfordCars | Aircraft | ImageNet | ADE20K | COCO(AP$^{bb}$) | COCO(AP$^{mk}$) |
| --- | --- | --- | --- | --- | --- | --- | --- | --- | --- | --- |
| SimMIM | 75.40±0.27 | 73.08±0.25 | 65.50±0.37 | 69.46±0.19 | 87.94±0.24 | 73.80±0.26 | 83.02±0.09 | 42.46±0.15 | 47.06±0.40 | 46.88±0.44 |
| MAE | 79.14±0.34 | 77.86±0.26 | 68.18±0.27 | 70.84±0.22 | 92.20±0.27 | 81.56±0.24 | 82.74±0.15 | 43.14±0.08 | 45.86±0.12 | 41.74±0.15 |
| DiffMAE | 78.50±0.32 | 76.22±0.22 | 68.66±0.23 | 64.08±0.14 | 90.46±0.27 | 79.74±0.18 | 82.16±0.28 | 42.66±0.10 | 44.18±0.24 | 38.64±0.21 |
| MaskDiT | 78.80±0.29 | 77.40±0.25 | 68.46±0.07 | 68.74±0.12 | 92.00±0.05 | 80.80±0.24 | 82.44±0.15 | 42.58±0.31 | 43.16±0.12 | 40.82±0.18 |
| Ours | **81.76**±0.36 | **81.16**±0.24 | **69.58**±0.15 | **72.26**±0.11 | **92.60**±0.24 | **84.42**±0.17 | **83.41**±0.13 | **43.54**±0.14 | **48.44**±0.38 | **47.90**±0.30 |

All experiments used ViT-B (Dosovitskiy et al., 2020) as the backbone architecture applying a unified 400-epoch training schedule. pretraining was performed on the ImageNet-1K (Deng et al., 2009) classification dataset, followed by fine-tuning on respective downstream task datasets.

## 5.2 MAIN RESULT

In Tab. 1, we evaluate the proposed method on diverse tasks, including fine-grained visual categorization (FGVC), image classification, semantic segmentation, object detection, and instance segmentation, each with task-specific datasets. FGVC datasets (CUB-200-2011 (Wah et al., 2011), NABirds (Van Horn et al., 2015), iNaturalist 2017 (Van Horn et al., 2017), iNaturalist 2018 (Van Horn et al., 2018), Stanford Cars (Krause et al., 2013), Aircraft (Maji et al., 2013)) demand detailed, fine-grained feature learning to distinguish visually similar classes. In contrast, standard recognition tasks (ImageNet (Deng et al., 2009)), semantic segmentation (ADE20K (Zhou et al., 2017)), object detection and instance segmentation (COCO (Lin et al., 2014)) emphasize broader spatial details at different levels of granularity.

To ensure the *statistical significance* of the results, we conducted 5 trials with different random seeds and included the mean and standard deviation for each method in the figure. In the graph, the bars represent the mean performance, while the lines indicate the standard deviation.

The proposed method (Ours) consistently outperforms representative MIM (Xie et al., 2022; He et al., 2022) and noise-based methods (Wei et al., 2023; Zheng et al., 2023) across tasks. In FGVC tasks, our method effectively captures high-frequency, localized features, as also shown in Fig. 1 and Fig. 2, surpassing comparison methods in accuracy. Even in standard recognition tasks, where spatial detail is key, our method shows favourable gains, highlighting our proposed design enhance transfer potential and capture diverse frequency information, as demonstrated in Fig. 1. These statistically significant improvements strongly validate the effectiveness and robustness of our approach over comparison methods.

**Ablation studies in the Appendix.** Comprehensive ablation studies are provided in the Appendix below and should be consulted for a complete understanding of our method. They cover various decoupling frameworks, time-embedding placement, longer pre-training schedules, evaluation on denoising task, and extended comparisons with related works.

## 6 CONCLUSION

We have investigated why existing noise-based C2R pretraining yields only limited gains on recognition tasks. Through systematic analysis, we proposed architectural guidelines that advocate encoder-style corruption, feature-level noise injection, and explicit disentanglement of masking and noising objectives. Our framework following these principles captures a broader frequency spectrum and achieves consistent improvements, surpassing both MIM and prior noise-based methods by significant margins across standard and fine-grained benchmarks. We believe that these findings highlight the importance of corruption design in self-supervised pretraining and open new directions to exploit generative principles in representation learning. Nevertheless, our analysis is currently limited to recognition tasks. It would be interesting to broaden the scope of our study to other applications.

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

# Appendix

## A  ABLATION STUDY ON DECOUPLING FRAMEWORKS

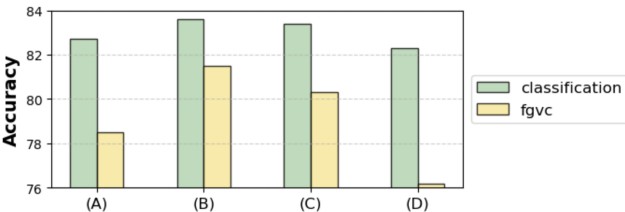

Figure 7: Comparison of four decoupling frameworks for combining de-noising (dn) and de-masking (dm) across encoder/decoder: (A) Enc-dn, Dec-dm; (B) Enc-dn+dm with disentanglement (proposed); (C) Enc-dm, Dec-dn; (D) Dec-dn+dm with disentanglement. Under matched pre-training and fine-tuning on ImageNet and FGVC, results suggest B > C > A > D, indicating a role for task placement. Encoder-centric placement with explicit separation tends to reduce cross-objective interference and yield more transferable features, while fully decoder-based training is weaker on average.

One key finding of this study is the critical role of disentangling de-noising and de-masking tasks within the encoder, as it is the component transferred during downstream fine-tuning. To maximize transfer learning potential, our framework explicitly separates these tasks through a disentanglement objective while keeping them within the encoder. To further investigate this, we evaluated *four decoupling frameworks* on fine-grained visual categorization (FGVC) (Wah et al., 2011) and image classification (Deng et al., 2009) tasks, varying the placement of de-noising and de-masking tasks across the encoder and decoder.

Specifically, we implemented and analyzed the following four configurations.

- **(A)** Encoder de-noises, decoder de-masks (Zheng et al., 2023).
- **(B)** Proposed framework: Encoder de-noises and de-masks with disentanglement loss, ensuring task separation.
- **(C)** Encoder de-masks, decoder de-noises.
- **(D)** Decoder de-noises and de-masks with disentanglement loss, fully shifting both tasks to the decoder.

The results in Fig. 7 suggest an ordering **(B)** > **(C)** > **(A)** > **(D)** under our protocol, pointing to a role for task placement. Framework (B) attains the highest mean score, indicating potential benefits when both objectives reside in the encoder with an explicit separation of responsibilities; a plausible explanation is reduced cross-objective interference while letting de-noising and mask reconstruction shape transferable features. Comparing (A) and (C) hints that de-masking inside the encoder can be more helpful than de-noising in our setup, possibly because mask reconstruction pressures features to encode part-level and semantic cues, whereas denoising can emphasize lower-level statistics. Framework (D) trails on most benchmarks, consistent with encoder supervision being more indirect when both objectives sit in the decoder. Overall, we read these trends as supportive of an encoder-centric placement with explicit separation, yielding semantically meaningful and transferable features across downstream tasks.

## B  ABLATION STUDY ON TIME-EMBEDDING

In denoising diffusion models, time-embedding plays a crucial role that encodes the temporal information associated with the noise levels introduced at various steps. Injecting time-embedding at appropriate points that align with the temporal dynamics of the noise is essential, as it enables the model to more effectively capture the relationship between the noise levels and the input features. Thus, for pre-training, we introduce time-embedding just before the block where noise is

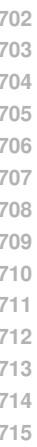

Figure 8: We evaluated the placement of time-embedding during fine-tuning on FGVC (Wah et al., 2011) and image classification tasks (Deng et al., 2009). Results show that the placement at block 2, consistent with pre-training, achieves the best performance by aligning temporal encoding with noise dynamics to retain fine-grained features.

added (block 2) to ensure the optimal alignment between the noise addition process and temporal information.

To evaluate how well the model utilizes fine-grained features aligned with temporal information for downstream tasks, we adjusted the placement of time-embedding during fine-tuning and assess its performance on FGVC (Wah et al., 2011) and image classification (Deng et al., 2009) tasks. Specifically, we tested time-embedding at four locations: initial embedding (block 0), immediately prior to noise addition (block 2), and post-noise addition blocks (block 4 and block 6).

In Fig. 8, the results show that placing time-embedding at block 2 achieves the best performance, followed by block 0, block 4, and block 6. This indicates that using the same time-embedding placement during fine-tuning as the pre-training (block 2) yields optimal results. The alignment of temporal encoding with noise dynamics is crucial for retaining the fine-grained features learned during pre-training.

The placement at block 0 achieves the second-highest performance, as injecting time-embedding at the initial stage allows the model to incorporate temporal information from the very beginning, allowing it to guide the extraction of features that are consistent with the progression of noise levels across the diffusion process. This observation aligns with findings reported in prior diffusion model studies (Ho et al., 2020; Nichol & Dhariwal, 2021), where early time-embedding helps initialize representations that remain consistent throughout the network. Blocks 4 and 6 perform worse as they occur after the noise has already been added and partially processed, making the temporal information less relevant to the feature refinement for downstream tasks. Late-stage time-embedding can introduce redundancy by repeating temporal information already captured in earlier layers, or fail to impact the already-learned representations effectively.

## C  FGVC EVALUATION ON LONGER PRE-TRAINING SCHEDULE

All experiments in the manuscript use a 400-epoch pre-training schedule. To assess schedule length effects, we also pre-train for 800 epochs and evaluate on FGVC (Wah et al., 2011). Figure 9 reports our method alongside representative MIM baselines—MAE (He et al., 2022) and SimMIM (Xie et al., 2022)—under both 400 and 800 epochs. Across methods, extending to 800 epochs yields consistently higher accuracy, but the improvements are modest relative to the additional compute. Notably, the relative ranking among methods is largely preserved, with no systematic cross-overs, suggesting that longer schedules primarily refine existing representations rather than alter inductive biases. Taken together, these results indicate that while longer schedules are beneficial, the cost–benefit trade-off is weak in this regime and does not change our main conclusions.

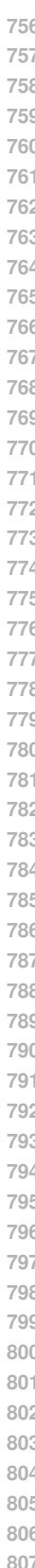

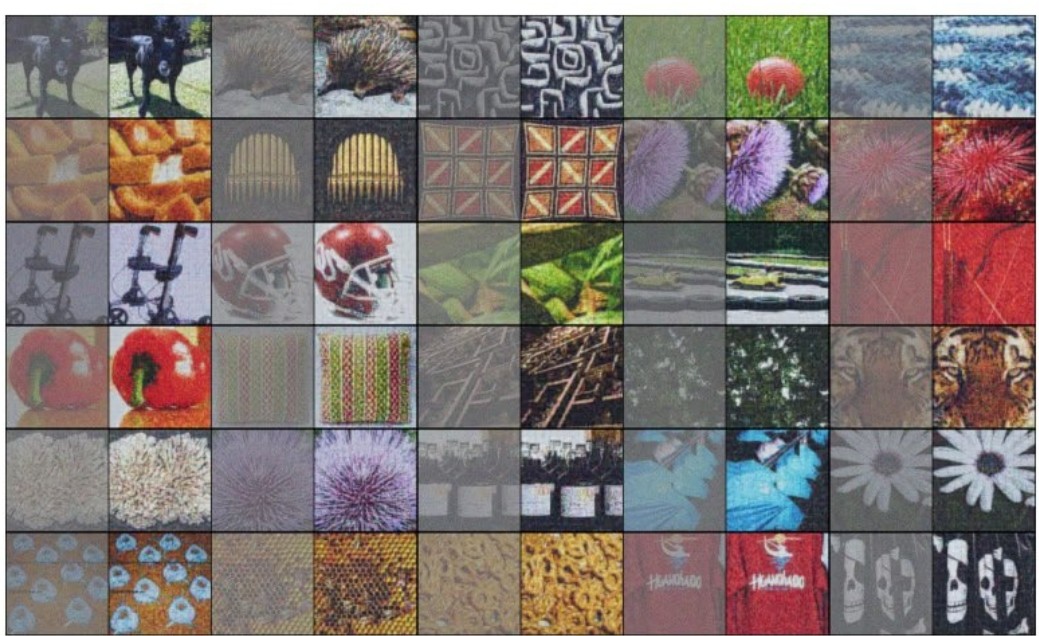

Figure 9: Effect of pre-training schedule length (400 vs. 800 epochs) on FGVC (Wah et al., 2011). We report our method and MIM baselines (MAE (He et al., 2022), SimMIM (Xie et al., 2022)). Accuracy increases at 800 epochs across methods, and rankings are largely preserved. Longer schedules appear to refine rather than change representations, yielding limited cost–benefit and leaving our conclusions unchanged.

## D  EVALUATION ON DENOISING TASK

Figure 10: We qualitatively evaluate our method on denoising tasks, demonstrating its ability to accurately reconstruct corrupted scenes by capturing fine-grained details and high-level semantics through effective de-noising and de-masking strategy. (Left: input images with applied noise. Right: Denoised outputs produced by the proposed model.)

The main manuscript evaluates a broad set of recognition tasks including FGVC, image classification, semantic segmentation, object detection, and instance segmentation. Since our approach builds

on denoising diffusion models, we hypothesize potential benefits on denoising-style downstreams as well.

To validate this, we provide qualitative results on denoising tasks over ImageNet validation benchmarks (Deng et al., 2009), as shown in Fig. 10. Our method demonstrates fair predictions even under heavy noise on the task that requires capturing both fine-grained textures and high-level semantics. This highlights how our model leverages noise-based learning to understand holistic scene representations, enabling it to also excel in generative tasks. The combined de-noising and de-masking framework further promotes the learning of semantic discriminability, enabling precise reconstruction of corrupted scenes.

## E   EXTENSION TO MULTIMODAL PRETRAINING

The principles proposed in this work—encoder-style corruption (P1), feature-level noise (P2), and explicit disentanglement (P3)—are fundamentally modality-agnostic. We hypothesized that these insights for optimizing Corruption-to-Reconstruction (C2R) are applicable to multimodal self-supervised learning frameworks. To validate this generalization capability, we implemented our C2R framework on top of established multimodal architectures: AIM-v2 (Fini et al., 2025) and CLIP (Radford et al., 2021).

The experiments utilized a ViT-L (Dosovitskiy et al., 2020) backbone for AIM-v2 (Fini et al., 2025) and a ViT-B (Dosovitskiy et al., 2020) backbone for CLIP (Radford et al., 2021). The results, summarized in Table 2, confirm that our principles generalize effectively to multimodal settings. Optimized corruption strategies enhance the joint representation quality, leading to consistent improvements over the baselines.

Table 2: Performance improvement when integrating our C2R principles into multimodal frameworks (ImageNet-1K Accuracy).

| Framework | Baseline | Baseline + Ours |
|---|---|---|
| AIM-v2 | 86.23 | **86.71** |
| CLIP | 83.35 | **84.83** |

## F   EVALUATION WITH CONTRASTIVE LEARNING AND OBJECTIVE CORRELATION

Our work primarily focuses on optimizing the C2R paradigm. However, we also investigated the relationship between our optimized C2R objectives and contrastive learning objectives.

**Correlation between Objectives.** MIM (de-masking) and Denoising primarily target local discrimination but operate on different frequency spectra (semantic structure vs. high-frequency details). Conversely, contrastive losses (Grill et al., 2020; Oord et al., 2018; Chen & He, 2021), focus on global instance discrimination. We hypothesized that these objectives target distinct aspects of representation learning and would thus be complementary.

**Empirical Evaluation.** To demonstrate this complementarity, we integrated a standard contrastive loss (Grill et al., 2020) into our C2R framework. All experiments utilized ViT-B Dosovitskiy et al. (2020) pretrained for 400 epochs. As shown in Table 3, integrating the contrastive loss improved our result (approx. +0.2%). The additive gains confirm that the improvements from our principled corruption design are distinct from, and complementary to, those emphasized by the contrastive loss. We also compare with other established contrastive MIM methods (Zhou et al., 2021; Yi et al., 2022; Mishra et al., 2022), demonstrating competitive performance.

## G   SYSTEMATIC COMPARISON OF EACH COMPONENT

To provide a consistent and directly comparable analysis of the contribution of each proposed principle, we provide a systematic, cumulative ablation study. We conducted these experiments under

Table 3: Comparison with Contrastive MIM methods and integration with contrastive loss (ViT-B, 400 epochs, ImageNet-1K Accuracy).

| Method | Accuracy |
|---|---|
| iBot (Zhou et al., 2021) | 81.05 |
| ConMIM (Yi et al., 2022) | 83.17 |
| CAN (CMA) (Mishra et al., 2022) | 83.49 |
| Ours | 83.41 |
| Ours + Contrastive Loss | **83.61** |

a unified setting (ViT-B (Dosovitskiy et al., 2020), 400 epochs) with a *fixed random seed* to ensure precise attribution of gains.

The results are presented in Table 4. This analysis demonstrates that Principle 1 (Encoder-style) provides a foundational improvement over the Decoder-style baseline, and the notable gains stem from Principle 2 (effective high-frequency capture via feature-level noise) and Principle 3 (reducing interference via disruption loss). P3 is crucial for unlocking the potential of combining P1 and P2 within the encoder architecture.

Table 4: Systematic cumulative ablation study of the proposed principles (ViT-B, 400 epochs, ImageNet-1K Accuracy).

| Method Configuration | Accuracy |
|---|---|
| Baseline (Decoder-style C2R) | 82.79 |
| + P1 (Encoder-style, Naive) | 83.12 |
| + P2 (Feature-level Noise, Blk 2) | 83.29 |
| + P3 (Disentanglement/Disruption) | **83.41** |

# H  COMPLEMENTARITY WITH RECENT MIM METHODS

Our study is analytical, focusing on foundational C2R paradigms (MAE He et al. (2022), Sim-MIM Xie et al. (2022), DiffMAE Wei et al. (2023), MaskDiT Zheng et al. (2023)) to derive general principles. A key finding is that these principles are complementary to specific architectural designs or masking strategies employed by existing MIM methods.

To validate this complementarity, we applied our C2R framework on top of several recent state-of-the-art MIM methods Hinojosa et al. (2024); Choi et al. (2024; 2025); Wang et al. (2023). All experiments utilized ViT-B pretrained for 400 epochs.

The results, summarized in Tab. 5, show consistent performance improvements across all baselines when our principles are integrated. This demonstrates the generality and effectiveness of our findings in enhancing representation learning within the broader MIM landscape.

Table 5: Demonstrating the complementarity of our C2R principles with recent MIM methods (ViT-B, 400 epochs, ImageNet-1K Accuracy).

| Method | Venue | Baseline | Baseline + Ours |
|---|---|---|---|
| ColorMAE (Hinojosa et al., 2024) | ECCV 2024 | 82.74 | 83.31 |
| MTO (Choi et al., 2024) | ECCV 2024 | 83.45 | 83.85 |
| SBAM (Choi et al., 2025) | ECCV 2024 | 83.65 | **84.09** |
| HPM (Wang et al., 2023) | CVPR 2023 | 83.63 | 83.83 |

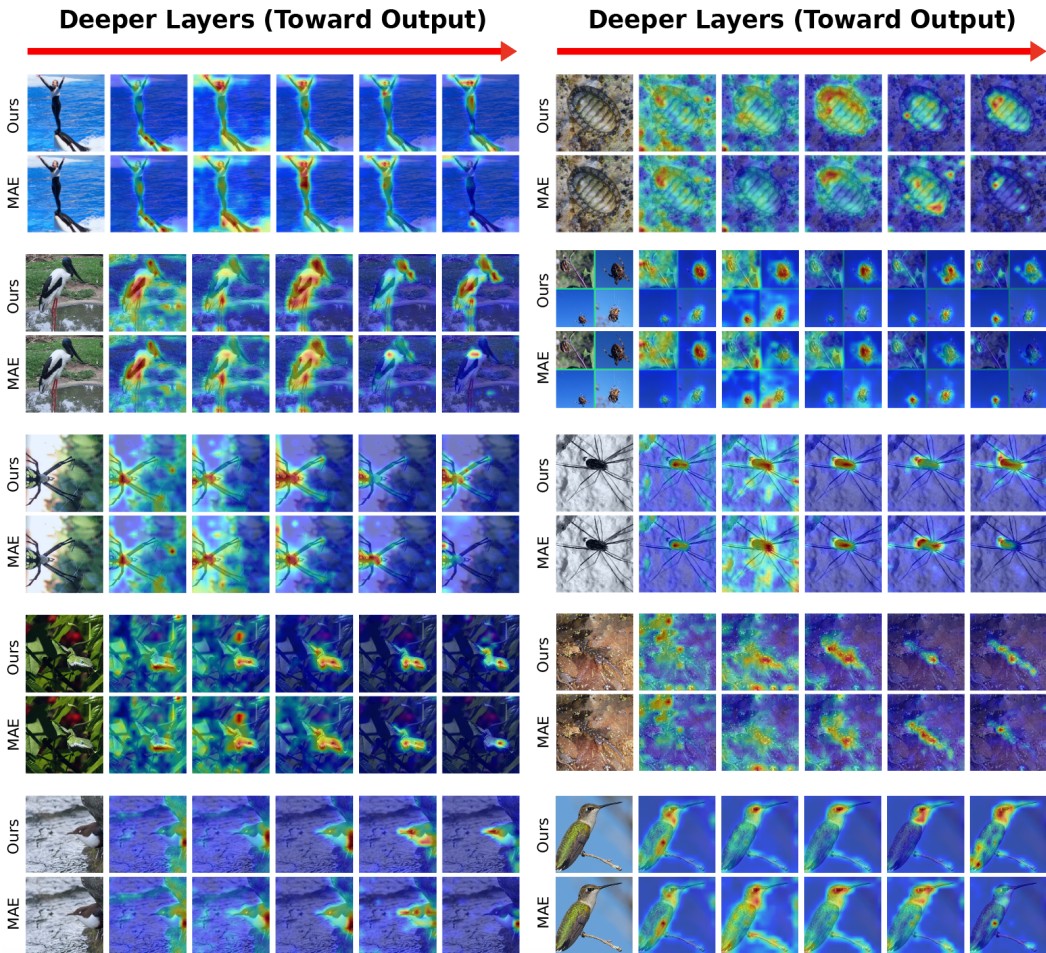

Figure 11: We present extensive visualizations comparing the layer-wise attention maps of a representative baseline (He et al., 2022) and the proposed approach. These visualizations demonstrate that our method leads to more comprehensive *latent feature* capture.

## I   VISUALIZATION OF LEARNED FEATURES

To provide qualitative insights into the representations learned by our framework, we include extensive visualizations comparing the layer-wise attention maps of a representative baseline (MAE) and the proposed approach in Fig. 11. These visualizations demonstrate that our method leads to more comprehensive capture of latent representation. Consistent with the quantitative analysis in Fig. 1 and the qualitative examples in Fig. 2, the attention mechanisms in our model focus simultaneously on high-level semantic regions and fine-grained details, resulting in richer and more transferable representations.

## J   COMPUTATIONAL COST ANALYSIS

We provide an analysis of the computational cost of our proposed method compared to representative MIM baselines. We report GFLOPS (for a standard 224x224 input) and the actual training time required for the 400-epoch pretraining schedule using ViT-B (Dosovitskiy et al., 2020) on our hardware configuration.

As shown in Table 6, our method adopts an Encoder-style architecture, similar to SimMIM, which processes all tokens in the encoder. While this results in higher GFLOPS compared to the Decoder-style MAE (which processes only visible tokens in the encoder), the actual training time differences

are relatively modest. In practice, runtime differences are often smaller than theoretical GFLOPS ratios due to implementation factors and hardware optimization. The computational cost of our method is comparable to SimMIM, and we believe the significant performance gains justify this trade-off.

Table 6: Computational cost analysis (ViT-B, 400 epochs).

| Method | Style | GFLOPs | Training Time |
|--------|-------|--------|---------------|
| MAE | Decoder-style | 11.67 | 51h |
| SimMIM | Encoder-style | 16.72 | 54h |
| Ours | Encoder-style | 16.94 | 55h |

## K  VISUALIZATION OF AFFINITY MATRIX CHANGES DUE TO DISRUPTION LOSS

To empirically validate the effect of the disruption loss ($\mathcal{L}_d$) on the attention mechanism, we analyzed the affinity matrices from the pretrained models, structured as described in Fig. 6(b). We visualized the affinity matrices with and without the application of $\mathcal{L}_d$.

In the baseline configuration (without $\mathcal{L}_d$), significant attention weights are observed in the cross-corruption quadrants (Q2: Masked-to-Noised, and Q3: Noised-to-Masked). This indicates potential interference between the two objectives. When the disruption loss is applied (Ours), these weights are effectively suppressed, and the weights in the quadrants Q1 and Q4 (Q1: Masked-to-Masked, and Q4: Noised-to-Noised) were increased. This confirms that $\mathcal{L}_d$ successfully mitigates the interference by recalibrating the attention distribution, thereby enforcing task disentanglement within the encoder.

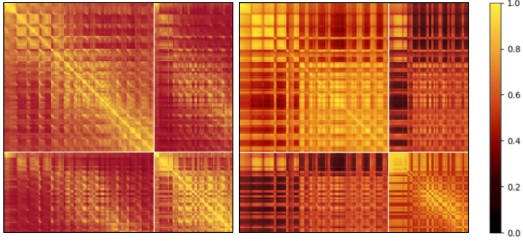

Figure 12: We visualized the affinity matrices with and without the application of $\mathcal{L}_d$. In the baseline (without $\mathcal{L}d$), strong attention appears in cross-corruption quadrants (Q2, Q3), indicating interference between masking and noising. With $\mathcal{L}d$, these interactions are suppressed while Q1 and Q4 intensify, showing that the loss recalibrates attention and effectively disentangles the two objectives.

## L  RELATIONSHIP BETWEEN MIM AND DIFFUSION MODELS

We clarify the relationship between Masked Image Modeling (MIM) and Denoising Diffusion Models (DDMs) within our Corruption-to-Reconstruction (C2R) framework, focusing on their integration for representation learning rather than generative modeling.

**(1) Intrinsic Similarity in Reconstruction Objectives.** Both MIM and DDM share a fundamental objective: recovering clean signals from corrupted inputs. MIM reconstructs from masking, while DDM reconstructs from additive noise. Due to this shared goal, it is feasible and efficient to utilize a unified architecture (encoder and decoder) to handle both reconstruction processes within a single-pass pretraining framework.

**(2) Differences in Corruption Application and Managing Interference.** While the objectives are similar, the nature of the corruptions (mask and noise) differs significantly. To effectively leverage these distinct mechanisms, we employ Conjunctive Corruption, applying them to separate token

sets: masking is applied to one subset of tokens, while diffusion-inspired noise (additive Gaussian noise with varying intensity) is applied to another subset. However, processing these heterogeneous corruptions simultaneously within a unified encoder still leads to interference, as the self-attention mechanism allows undesirable interaction between the masked and noised tokens.

It is precisely to mitigate this interference that we introduced the disruption loss. This loss is specifically designed to harmonize the training signals derived from the distinct corruption processes, addressing their inherent differences. This explicitly suppresses the cross-attention between the two token types, thereby effectively disentangling the MIM and DDM objectives.

## M  SCALABILITY ANALYSIS

The primary experiments in the manuscript utilized a ViT-B (Dosovitskiy et al., 2020) architecture with a 400-epoch pretraining schedule. This standard duration is widely accepted in recent MIM literature (e.g., MTO (Choi et al., 2024), SBAM (Choi et al., 2025), ColorMAE Hinojosa et al. (2024), ConMIM (Yi et al., 2022), CAE (Chen et al., 2024a), BootMAE (Dong et al., 2022)) as sufficient for convergence while balancing computational constraints. To address the scalability of our proposed principles, we conducted further experiments with larger models (ViT-L) and longer training schedules (800 and 1600 epochs).

The results are summarized in Table 7. Our method demonstrates effective scaling across all configurations. When scaling to ViT-L (400 epochs), our method achieves 84.84%, outperforming MAE and SimMIM. With longer schedules (1600 epochs, ViT-B), our method reaches 84.28%, maintaining a consistent advantage over the baselines. Furthermore, combining a larger model and longer schedule (ViT-L, 800 epochs) yields the highest performance of 85.24%. These results confirm that the benefits of our optimized C2R framework persist and amplify at scale.

Table 7: Scalability analysis with larger models and longer pretraining schedules (ImageNet-1K Accuracy).

| Architecture | Epochs | MAE | SimMIM | Ours |
|---|---|---|---|---|
| *Larger Models* | | | | |
| ViT-L | 400 | 84.17 | 84.03 | 84.84 |
| *Longer Schedules* | | | | |
| ViT-B | 800 | 83.31 | 83.64 | 83.93 |
| ViT-B | 1600 | 83.56 | 83.88 | 84.28 |
| *Larger Models and Longer Schedules* | | | | |
| ViT-L | 800 | 84.75 | 84.36 | **85.24** |

## N  IMPLEMENTATION DETAILS

This section outlines the implementation details. Tab. 8 and 9 summarize the specific settings and configurations used for our experiments. To maintain reproducibility and transparency, all codes are included in the Supplementary Material to support implementation clarity.

Table 8: Pre-Training Settings.

| Setting | Ours |
|---|---|
| Optimizer | AdamW |
| Base Learning Rate | 1e-4 |
| Weight Decay | 0.05 |
| Optimizer Momentum | $\beta_1 = 0.9, \beta_2 = 0.999$ |
| Learning Rate Schedule | Multi-Step (gamma=0.1, steps=300) |
| Batch Size | 4096 |

Table 9: Fine-Tuning Settings.

| Setting | Ours |
| --- | --- |
| Optimizer | AdamW |
| Base Learning Rate | 1.25e-3 |
| Weight Decay | 0.05 |
| Layer Decay | 0.65 |
| Optimizer Momentum | $\beta_1 = 0.9, \beta_2 = 0.999$ |
| Learning Rate Schedule | Cosine Decay |
| Drop Path | 0.1 |
| Batch Size | 4096 |
| Warmup Epoch | 20 |
| Training Epoch | 100 |
| RandAug | RandAug(9, 0.5) |
| Mixup | 0.8 |
| Cutmix | 1.0 |
| Label Smoothing | 0.1 |
| Random Erasing | 0.25 |

**Final Training Objective.** The final training objective combines the reconstruction losses for both masked and noised tokens with the disruption loss designed to disentangle their optimization trajectories. The total loss function $\mathcal{L}_{Total}$ is formulated as:

$$\mathcal{L}_{Total} = \mathcal{L}_{MIM} + \mathcal{L}_{Denoise} + \lambda\mathcal{L}_d, \qquad (11)$$

The hyperparameter $\lambda$ is set to 0.1. It is important to note that $\lambda$ does not require delicate tuning; it acts as a regularization coefficient that stabilizes training by aligning the magnitudes of the entropy-based disruption loss relative to the reconstruction losses.

## O   THE USE OF LLMS

LLMs were used only for minor language improvements. They were not involved in the conception of the research, experiments, analysis, interpretation, or drafting.

