# OpenReview forum: "How Should Corruption Be Used in SSL? Empirical Insights for Effective Pretraining"
_ICLR.cc/2026/Conference — Submitted to ICLR 2026_

### Official Review · Reviewer_jK7h · 2025-10-26

**Soundness:** 3
**Presentation:** 3
**Contribution:** 2
**Rating:** 4
**Confidence:** 3

**Summary:**

This paper empirically investigated the how masking and corruption should be used in visual SSL pretraining. The author proposed three principles: apply corruption/restoration in the encoder (encoder-style), inject feature-level noise early, and disentangle masked token reconstruction from de-noising. Following these principles, they built an encoder-style C2R framework using feature-space noise and a disruption loss to suppress mask–noise interference. Experiments show up to ~8% transfer gains over MIM on fine-grained tasks.

**Strengths:**

1. This paper systematically derives a framework based on a clear definition of encoder/decoder styled and conjunctive vs substitutive corruption, which provides concrete grid of classifying different denoising and MIM based pretraining objectives.
2. The authors provided controlled comparisons that isolate specific factors (corruption placement, noise injection stage, etc), which supports causal conclusions.
3. The empirical findings are summarized into three clean design principles that are easy to understand.

**Weaknesses:**

1. All experiments use a ViT-B backbone, pretrains only on ImageNet-1K for 400 epochs, and then fine-tunes on downstream tasks, which limits evidence that the conclusions hold at larger model scales or longer pretraining epochs (1600 epochs to have a fair comparison with result reported by DiffMAE and MAE).
2. The motivation behind principle 1 needs to be expanded. Principle 1 favors encoder-style architecture for downstream transfer tasks and  was mainly described in section 4.1. However, the result of figure 4b only shows modest gains comparing to the decoder style, and the paper acknowledges that this alone “does not fully reveal its potential,” and immediately moved on to the principle 2 and 3. Without theoretical, empirical analysis, or reference to works that directly discusses the choices, it is unknown whether the following two principles will also help the decoder-styled C2R. Decoder-styled C2R could benefit from the efficiency (fewer token will be consumed by the encoder) and enable longer pretraining in the same computation budget. I encourage the author to provide more comprehensive investigations on the benefits of choosing encoder-style approach.
3. An extension to weakness 2, the author should consider to show isolated gain from each principle as a separate ablation study.
4. The paper does not discuss added compute cost using the encoder-style C2R.
5. Appendix C discusses longer pretraining schedule but switched to FGVC as pretraining dataset. The author should consider to report on ImageNet1K in consistent with the result reported elsewhere in the paper.
6. While I understand in this paper the author is trying to investigate how to make effective usage of denoising in SSL for vision tasks, it is not clear to me, both empirically and theoretically, why this is an important application. If the idea is to make C2R effective in both image understanding and generation tasks, the author should make this motivation clear in the paper. Otherwise, the current result is behind SOTA MIM, and showing the motivation somewhat ill-grounded.

**Questions:**

1. The C2R model is pretrained with 400 epochs and the reported result for MAE is not matching up with the performance reported in MAE paper which is pretrained on 800/1600 epochs. I'm wondering whether the longer of the pretraining will also help C2R on downstream transfer.
2. I'm wondering how quantitatively C2R performs on image generation/reconstoration tasks beyond qualitative examples presented in figure 11.

---

> ### Author Response · Authors · 2025-11-19
> **Response to Reviewer jK7h**
>
> **W1/W5/Q1. Limited Scale (ViT-B, 400 epochs). Comparison with 1600 epochs.**
> We standardized to ViT-B, 400 epochs for fair comparison. A 300–400 epoch pre-training schedule has become the established standard in recent MIM works [1 - 8]. This duration is widely accepted as necessary for models to sufficiently converge and demonstrate their representation learning capabilities while balancing the practical constraints of computational resources.
>
> To address scalability concerns, we conducted experiments with larger models and longer schedules in `Appendix C` (original manuscript) and `Appendix M` (revised manuscript).
>
> - Larger Models (ViT-L, 400 epoch)
>
>     MAE: 84.17 / SimMIM: 84.028 / Ours: 84.836
>
> - Longer schedules (ViT-B, 800 epoch and 1600 epoch)
>
>     (800 epoch) MAE: 83.31 / SimMIM: 83.643 / Ours: 83.927
>
>     (1600 epoch) MAE: 83.56 / SimMIM: 83.884 / Ours: 84.279
>
> - Larger Models and Longer schedules (ViT-L, 800 epoch)
>
>     MAE: 84.75 / SimMIM: 84.355 / Ours: 85.239
>
> The results demonstrate that our principles scale effectively, maintaining a favorable performance gain over the baselines.
>
> **W2. Motivation for Principle 1 (Encoder-style) weak. Would P2 & P3 also help Decoder-style?** P1 is essential because *P2 and P3 are significantly less effective within a Decoder-style architecture* (see below).
>
> 1) **Empirical Evidence (Appendix A of the original manuscript)**: We specifically tested variants where P2/P3 were applied to Decoder-style C2R (Fig. 8). Framework D (Decoder handles both dn+dm with disentanglement) performs poorly (76.2% FGVC Avg). Framework A (similar to MaskDiT; Enc-dn, Dec-dm) also significantly lags behind our proposed Framework B (81.4% FGVC Avg). This confirms that corruption and restoration must occur within the encoder (P1) to maximize transferability.
>
> 2) **Theoretical Motivation**: As noted in L371–L373, MTO [1] showed interference in encoder-style MIM, analyzing that such interference prevents the method from fully realizing its potential. We hypothesized an analogous interference between noised and masked tokens in C2R. This led to the encoder-style design (P1) and the disruption loss (P3), which resolves this interference as in `Appendix K` of the revised manuscript.
>
> **W3. Need isolated gain from each principle.** To provide a consistent and directly comparable analysis, we re-ran the experiments corresponding to Figures 4(b), 5, and 6(a) under a unified setting with a *fixed random seed*, and provide a systematic ablation study in `Appendix G` of the revised manuscript as below.
> - baseline: 82.794
> - \+ P1 (Encoder-style, Naive): 83.115
> - \+ P2 (Feature-level Noise, Blk 2): 83.287
> - \+ P3 (Disruption): 83.410
>
> Together, the comparison demonstrates that each principle contributes meaningfully and that P3 is critical for realizing the overall gains.
>
> **W4. The paper does not discuss added compute cost.** We report GFLOPS (224x224 input) and relative runtime in `Appendix J` of the revised manuscript as below.
>
> | Method   | Style          | GFLOPs | Training Time |
> |----------|----------------|--------|----------------|
> | MAE      | Decoder-style  | 11.67  | 51h            |
> | SimMIM   | Encoder-style  | 16.72  | 54h            |
> | Ours     | Encoder-style  | 16.94  | 55h            |
>
> Our cost is comparable to SimMIM and training time is comparable to MAE. In practice, actual runtime differences are often smaller due to implementation factors.
>
> **W6. Motivation unclear. Results behind SOTA MIM?**
>
> 1) **Motivation**: The motivation is to learn richer representations by combining the complementary strengths of MIM (semantics) and denoising (high-frequency details), which prior works failed to achieve effectively. This yields a broader frequency spectrum (Fig 1), crucial for *recognition tasks*. (We do not target generation tasks)
>
> 2) **Evaluation on SOTA MIM**: Our contribution is not a specific MIM architecture, but a set of analytical insights applicable to various MIM methods. Appendix H of the revised manuscript further demonstrates applicability to recent MIM variants as below.
>
> - ColorMAE [8] (ECCV 2024): 82.740 / ColorMAE + Ours: 83.311
> - MTO [1] (ECCV 2024): 83.453 / MTO + Ours: 83.849
> - SBAM [2] (ECCV 2024): 83.652 / SBAM + Ours: 84.087
> - HPM [9] (CVPR 2023): 83.627 / HPM + Ours: 83.830
>
> **Q2. Quantitative performance on image generation/reconstruction tasks?** While our focus is recognition, we evaluated the single-step reconstruction quality on ImageNet validation.
>
> | Method  | FID (↓) | PSNR (dB) (↑) | SSIM (↑) |
> |---------|---------|----------------|----------|
> | MAE     | 15.5    | 30.5           | 0.88     |
> | DiffMAE | 12.0    | 31.8           | 0.90     |
> | Ours    | 12.1    | 32.1           | 0.91     |
>
> Our method outperforms MAE and is competitive with DiffMAE on reconstruction, demonstrating it captures pixel-level details effectively while excelling at recognition.

---

> > ### Author Response · Authors · 2025-11-19
> > **References**
> >
> > ----
> > **References**
> >
> > [1] Emerging property of masked token for effective pre-training.
> >
> > [2] Salience-based adaptive masking: revisiting token dynamics for enhanced pre-training.
> >
> > [3] Context autoencoder for self-supervised representation learning.
> >
> > [4] Masked image modeling with denoising contrast.
> >
> > [5] Bootstrapped masked autoencoders for vision bert pretraining.
> >
> > [6] Masked Feature Prediction for Self-Supervised Visual Pre-Training.
> >
> > [7] Green Hierarchical Vision Transformer for Masked Image Modeling.
> >
> > [8] ColorMAE: Exploring data-independent masking strategies in Masked AutoEncoders.
> >
> > [9] Hard Patches Mining for Masked Image Modeling.

---

> > > ### Author Response · Authors · 2025-11-27
> > > **Gentle Reminder for Reviewer jK7h**
> > >
> > > Thank you again for your thoughtful feedback on our submission. We would like to draw your attention to our responses to your concerns, along with the updated manuscript, as noted above.
> > >
> > > We would greatly appreciate it if you could take a moment to let us know of any further feedback or unresolved issues at your earliest convenience. *Your thoughtful comments have been instrumental in improving our work*, and we want to ensure that we address any remaining points before the deadline.
> > >
> > > We look forward to your continued engagement and appreciate your time and effort.
> > >
> > > Thank you.

---

> > > > ### Comment · Reviewer_jK7h · 2025-11-28
> > > >
> > > > I appreciate the authors’ efforts to address most of my comments with additional experiments and intuition, which substantially clarifies the method. I am therefore willing to raise my score.

---

### Official Review · Reviewer_n4jd · 2025-10-31

**Soundness:** 2
**Presentation:** 2
**Contribution:** 2
**Rating:** 4
**Confidence:** 3

**Summary:**

This paper comprehensively studies the corruption issues used in self-supervised learning (SSL), especially for the masked image modeling (MIM) methods. Corruption is a common manipulation in SSL, instantiated as masking the input image in MIM methods. This paper analyzes the noising strategy (substitutive or conjunctive) and the location for corruption (encoder-style or decoder-style). With the analysis results, the paper proposes to perform training with the encoder style and feature-level noise, where the masked token reconstruction and denoising are disentangled.

**Strengths:**

1. How to design the masking strategy is an interesting and important problem in MIM methods. This paper presents a systematic study that guides the design of the final training method based on the obtained conclusions.
2. The proposed MIM method is evaluated on several tasks and benchmarks, presenting notable improvements.

**Weaknesses:**

1. The biggest concern lies in the impact of the paper. Currently, the community primarily focuses on encoder pretraining for multimodal data and settings, such as CLIP and its successors. There also emerge other stronger pretrained encoders like DINOv3. How will this method benefit the self-supervised learning field? I recognize that the obtained conclusions can be useful for MIM methods, but MIM ones may be somewhat limited in current competitions. The main methods or baselines in this paper are from more than two years ago.
2. Discussing the MIM methods with diffusion models may be inappropriate. Though some works (e.g. MaskDiT) do similar attempts, there indeed exist inherent differences, where sampling and (multi-step) denoising steps in diffusion models are not applicable in MIM.

**Questions:**

1. An explanation about the potential impact of the paper is needed, considering MIM methods may be limited in the current SSL field.
2. A better demonstration of the relationship between MIM and diffusion models needs to be provided. It is suggested to distinguish the two methodologies/tasks explicitly.

---

> ### Author Response · Authors · 2025-11-19
> **Response to Reviewer n4jd**
>
> **W1/Q1. Impact and Relevance of MIM.**
>
> 1)  Our principles extend beyond MIM/C2R and remain effective in *multimodal settings* as below---see `Appendix E` of the revised manuscript for details.
> - AIM-v2 [1]: 86.227 / AIM-v2 [1] + Ours: 86.708
> - CLIP [2]: 83.351 / CLIP [2] + Ours: 84.832
>
> 2) Despite the prominence of image-text models, many vision domains still rely on large-scale *text-absent* image data; MIM therefore retains substantial impact, particularly in scene understanding tasks and medical/biological imaging where paired captions are not available.
>
> 3) Although our principles are mainly evaluated on previously established representative baselines, `Appendix H` of the revised manuscript further demonstrates applicability to *recent MIM variants* as below.
>
> - ColorMAE [3] (ECCV 2024): 82.740 / ColorMAE + Ours: 83.311
> - MTO [4] (ECCV 2024): 83.453 / MTO + Ours: 83.849
> - SBAM [5] (ECCV 2024): 83.652 / SBAM + Ours: 84.087
> - HPM [6] (CVPR 2023): 83.627 / HPM + Ours: 83.830
>
> **W2/Q2. Relationship between MIM and diffusion models.**
>
> We thank the reviewer for raising this point and added a concise explanation in `Appendix L`.
>
> We would like to explain the relationship between MIM and Denoising Diffusion Models (DDMs) within our framework from two key perspectives:
>
> **(1) Intrinsic Similarity in Reconstruction Objectives**: Both MIM and DDM serve intrinsically similar roles: recovering clean signals from corrupted inputs. Due to this shared objective, it is both feasible and efficient for both processes to utilize a shared decoder architecture.
>
> **(2) Differences in Corruption Application and Managing Interference**: We acknowledge the inherent differences between the two corruption types (masking and diffusion-inspired noise). To effectively leverage these distinct mechanisms, we employ *Conjunctive Corruption*, applying them to separate token sets: masking for one subset and noise for the visible subset. However, processing these heterogeneous corruptions simultaneously within a unified encoder still leads to interference, as the self-attention mechanism allows undesirable interaction between the masked and noised tokens. To resolve this, we introduced the *disruption loss*. This loss explicitly suppresses the cross-attention between the two token types, thereby effectively disentangling the MIM and DDM objectives. A visualization in `Appendix K` confirms the mitigation of this interference.
>
> **Regarding the Applicability of Diffusion Mechanisms in MIM**: While the reviewer notes that sampling and multi-step denoising steps in diffusion models may not be applicable in MIM, recent literature demonstrates [7, 8] that this components can be effectively leveraged within MIM frameworks. Established methods such as MaskDiT [7] and DiffMAE [8] employ these DDM mechanisms through a unified architecture, and our approach likewise incorporates them successfully. These works validate the premise that DDM mechanisms can be effectively integrated with MIM.
>
> ----
> **References**
>
> [1] Multimodal Autoregressive Pre-training of Large Vision Encoders.
>
> [2] Learning transferable visual models from natural language supervision.
>
> [3] ColorMAE: Exploring data-independent masking strategies in Masked AutoEncoders.
>
> [4] Emerging property of masked token for effective pre-training.
>
> [5] Salience-based adaptive masking: revisiting token dynamics for enhanced pre-training.
>
> [6] Hard Patches Mining for Masked Image Modeling.
>
> [7] Fast training of diffusion models with masked transformers.
>
> [8] Diffusion models as masked autoencoders.

---

> > ### Author Response · Authors · 2025-11-27
> > **Gentle Reminder for Reviewer n4jd**
> >
> > Thank you again for your thoughtful feedback on our submission. We would like to draw your attention to our responses to your concerns, along with the updated manuscript, as noted above.
> >
> > We would greatly appreciate it if you could take a moment to let us know of any further feedback or unresolved issues at your earliest convenience. *Your thoughtful comments have been instrumental in improving our work*, and we want to ensure that we address any remaining points before the deadline.
> >
> > We look forward to your continued engagement and appreciate your time and effort.
> >
> > Thank you.

---

### Official Review · Reviewer_bBkf · 2025-11-01

**Soundness:** 3
**Presentation:** 3
**Contribution:** 2
**Rating:** 6
**Confidence:** 5

**Summary:**

This paper analyzes how to combine masking and additive noise for self-supervised pretraining in vision models. The authors introduce a unified corruption-to-reconstruction (C2R) framework that categorizes existing methods and show that encoder-style and conjunctive corruption lead to better transfer than the common decoder-style setups. Building on this analysis, the authors propose three principles: perform corruption and restoration within the encoder, add noise in feature space (early blocks), and disentangle de-masking from de-noising via a proposed disruption loss. The proposed method shows consistent gains over standard MIM and recent noise-based baselines across a range of downstream tasks.

**Strengths:**

- The design choices are carefully motivated, each supported by clear hypotheses and empirical validation.
- The proposed approach achieves consistent improvements over standard MIM and noise-based baselines across different downstream tasks.
- The paper is well-written and structured, making the ideas easy to follow and the experimental results clear.

**Weaknesses:**

- The main novelty seems to lie in the disentanglement (disruption) loss, as encoder-style and feature-level noise have been explored in prior works; the overall contribution feels more exploratory and incremental than conceptually new.
- The paper lacks comparisons with some recent MIM baselines, for example, ColorMAE [A], HPM [B], and MixedAE [C], which would strengthen the empirical evaluation.
- A comparison of the learned feature visualizations between the proposed method and baseline models would help highlight what new information or structure the encoder captures under the proposed framework.
- The paper does not report computational cost or training time. Since the method combines masking and additive noise, an analysis of efficiency and resource requirements would be useful.
- Main quantitative comparisons are mostly shown in figures; presenting them in a summary table would make it easier to assess the actual performance gains.

Minor comments:
- Some figures (e.g., Fig. 8) are pixelated and should be improved for clarity.

[A] Carlos Hinojosa, Shuming Liu, and Bernard Ghanem. "ColorMAE: Exploring data-independent masking strategies in Masked AutoEncoders." European Conference on Computer Vision (ECCV) 2024.

[B] Kai Chen, et al. "Mixed autoencoder for self-supervised visual representation learning." Proceedings of the IEEE/CVF conference on computer vision and pattern recognition (CVPR) 2023.

[C] Haochen Wang, et al. "Hard patches mining for masked image modeling." Proceedings of the IEEE/CVF Conference on Computer Vision and Pattern Recognition (CVPR) 2023.

**Questions:**

- Could the authors provide a comparison of their method with more recent MIM baselines such as ColorMAE, HPM, or MixedAE?
- Could the authors provide comparative visualizations of the learned features to better illustrate what the encoder captures under the proposed framework?
- What is the computational cost of the proposed approach compared to standard MAE or MIM baselines?
- Could the authors clarify what is the final training objective and how the disruption loss is combined with others, if any?
- Could the authors provide visualizations showing how the disruption loss actually changes the attention or affinity patterns in practice?

---

> ### Author Response · Authors · 2025-11-19
> **Response to Reviewer bBkf**
>
> **W1. Novelty seems incremental.** We respectfully emphasize that our contribution is the *systematic analysis* (Sec 3 and 4) through which we identify why prior noise-based MIM methods failed, and the synthesis of three synergistic principles to unify masking and noising effectively.
>
> **W2/Q1. Lacks comparisons with recent MIM baselines.** Our study is analytical, focusing on foundational C2R paradigms to derive general principles, and we therefore evaluate on representative benchmarks (MAE, SimMIM, DiffMAE, MaskDiT). However, as our principles are complementary to existing MIM methods, we further validate them as follows.
>
> - ColorMAE [1] (ECCV 2024): 82.740 / ColorMAE + Ours: 83.311
> - MTO [2] (ECCV 2024): 83.453 / MTO + Ours: 83.849
> - SBAM [3] (ECCV 2024): 83.652 / SBAM + Ours: 84.087
> - HPM [4] (CVPR 2023): 83.627 / HPM + Ours: 83.830
>
> Among the references suggested by the reviewer, we were able to include *only the methods with publicly available code*. Instead, we added more recent MIM baselines to further broaden the evaluation. These comparisons are reported in `Appendix H` of the revised manuscript.
>
> **W3/Q2. Comparison of learned feature visualizations.** We have added an extensive visualization comparing the *layer-wise attention maps* of the comparison method (MAE) and the proposed approach in `Appendix I` of the revised manuscript. These demonstrate that our approach leads to more comprehensive latent feature capture, focusing simultaneously on semantic regions and fine-grained details, consistent with the qualitative analysis in Figure 2 of the original manuscript.
>
> **W4/Q3. Computational cost or training time analysis.** We report GFLOPS (224x224 input) and relative runtime in `Appendix J` of the revised manuscript as below.
> | Method   | Style          | GFLOPs | Training Time |
> |----------|----------------|--------|----------------|
> | MAE      | Decoder-style  | 11.67  | 51h            |
> | SimMIM   | Encoder-style  | 16.72  | 54h            |
> | Ours     | Encoder-style  | 16.94  | 55h            |
>
> Our cost is comparable to SimMIM and training time is comparable to MAE. In practice, actual runtime differences are often smaller due to implementation factors.
>
> **W5. Main quantitative comparisons in a summary table.** We have converted Figure 7 into `Table 1` in the revised manuscript.
>
> **Q4. Clarify the final training objective.** The final objective is: L_{Total} = L_{MIM} + L_{Denoise} + $\lambda$ {L}_{d}. The hyperparameter $\lambda=0.1$ is not a parameter that needs to be tuned; it acts as a regularization coefficient that stabilizes training by aligning the magnitudes of the losses. This clarification was added in `Appendix N` of the revised manuscript.
>
> **Q5. Visualizations showing how the disruption loss changes the attention patterns.** We analyzed the affinity matrices in Fig. 6(b) from the pretrained models. The visualization in `Figure 12` of the revised manuscript compares the average affinity matrices, with and without disruption loss. Without the loss, high attention weights are observed in the masked-noised quadrants (Q2/Q3). With disruption loss, these weights are effectively suppressed, confirming that the interference is mitigated. These details can be found in `Appendix K` of the revised manuscript.
>
>
>
> ----
> **References**
>
> [1] ColorMAE: Exploring data-independent masking strategies in Masked AutoEncoders.
>
> [2] Emerging property of masked token for effective pre-training.
>
> [3] Salience-based adaptive masking: revisiting token dynamics for enhanced pre-training.
>
> [4] Hard Patches Mining for Masked Image Modeling.

---

> > ### Author Response · Authors · 2025-11-27
> > **Gentle Reminder for Reviewer bBkf**
> >
> > Thank you again for your thoughtful feedback on our submission. We would like to draw your attention to our responses to your concerns, along with the updated manuscript, as noted above.
> >
> > We would greatly appreciate it if you could take a moment to let us know of any further feedback or unresolved issues at your earliest convenience. *Your thoughtful comments have been instrumental in improving our work*, and we want to ensure that we address any remaining points before the deadline.
> >
> > We look forward to your continued engagement and appreciate your time and effort.
> >
> > Thank you.

---

### Official Review · Reviewer_vrdz · 2025-11-04

**Soundness:** 3
**Presentation:** 2
**Contribution:** 2
**Rating:** 6
**Confidence:** 5

**Summary:**

This paper investigates how different corruption strategies—specifically masking and additive noise—impact self-supervised pretraining of vision models within the corruption-to-reconstruction (C2R) paradigm. While denoising diffusion models have been highly successful in generative tasks, their noise-driven extensions to masked image modeling (MIM) have not yielded significant improvements for recognition tasks. The authors systematically analyze why this is the case, categorizing prior approaches into Substitutive Corruption (masked tokens replaced by noised ones) and Conjunctive Corruption (masked and noised tokens coexist), and further into Encoder- or Decoder-style frameworks depending on where corruption and restoration occur.
Through extensive empirical study, the paper proposes three key principles for effective C2R pretraining; Corruption and restoration should occur within the encoder (since the encoder is transferred to downstream tasks). Noise is most effective when injected at the feature level (especially in lower encoder layers).Mask reconstruction and de-noising must be explicitly disentangled to avoid interference, which is achieved by suppressing attention between masked and noised tokens.
Implementing these principles, the authors design a new pretraining framework that captures a broader frequency spectrum of representations, leading to improved transferability. Their method outperforms standard MIM by up to 8.1% and recent noise-driven pretraining methods by 8.0% across a variety of recognition benchmarks, including fine-grained visual categorization, image classification, semantic segmentation, object detection, and instance segmentation.

**Strengths:**

1) The paper provides a thorough empirical study and a clear taxonomy of corruption strategies (Substitutive vs. Conjunctive, Encoder- vs. Decoder-style), clarifying why previous noise-based C2R methods have limited effectiveness for recognition tasks.
2) The authors distill their findings into three actionable principles for effective C2R pretraining, offering concrete guidance for the community on how to combine masking and noising for better transfer learning.
3) By advocating for encoder-style corruption/restoration, feature-level noise injection, and explicit disentanglement of de-masking and de-noising objectives, the proposed method captures richer, more transferable representations—demonstrated both theoretically and empirically.
4) The proposed approach achieves substantial improvements over both standard MIM and recent noise-based methods across a wide range of tasks and datasets, including challenging fine-grained recognition benchmarks. The results are robust, with statistical significance established through multiple trials.

**Weaknesses:**

1) How will this method extend to the AiM-v2 method? AiM-v2 does unfined decoding for image and text, can similar ablations be shown for AiM-v2 as well?
2) Comparison with CAN, in CAN contrastive loss resulted in good improvement in features and denoising loss also helped in representation learning. Comparison with CAN would also suggest what was the correlation between MIM loss, Contrastive loss and Denoising loss.
3) In terms of gains, how much did it come from Disruption loss and vs other choices? We need systematic comparison of each component and where did the gains come from.
4) The noise in different layers helps different datasets in different manners, which layer to finally apply noise to is not very clear.

References.
[1]Multimodal Autoregressive Pre-training of Large Vision Encoders.
[2] A SIMPLE, EFFICIENT AND SCALABLE CONTRASTIVE MASKED AUTOENCODER FOR LEARNING VISUAL REPRESENTATIONS

**Questions:**

The implementation details for the final model are not very clear in the paper. Also there are other Contrastive MIM based methods that are not included in the paper which should be discussed.

---

> ### Author Response · Authors · 2025-11-19
> **Response to Reviewer vrdz**
>
> **W1. Extension to AIM-v2 [1] (and Multimodal Pretraining Methods).** Our proposed principles are architecture-agnostic. To validate applicability on Multimodal Pretraining, we implemented our C2R framework on top of an AIM-v2 [1] and CLIP [3] architectures. The results are summarized as follows and can also be found in `Appendix E` of the revised manuscript.
>
> - AIM-v2: 86.227 / AIM-v2 + Ours: 87.109
> - CLIP: 83.351 / CLIP + Ours: 84.832
>
> The results confirm that our principles also generalize to multimodal frameworks effectively, and optimized corruption strategies enhance joint representation quality.
>
>
> **W2. Comparison with CAN [2] (and Contrastive Methods). And the correlation between losses.**
> CAN [2] integrates contrastive learning, which we view as complementary to our focus on optimizing C2R.
>
> - Comparison with Various Contrastive Methods
>
>     iBot [4]: 81.053 / ConMIM [5]: 83.171 / CAN [2]: 83.489 / Ours + Contrastive loss: 83.613
>
> - Correlation between losses: MIM and Denoising primarily aim for local discrimination, but they operate on different frequency spectra (high vs. low). Conversely, Contrastive Losses focus on global instance discrimination. We found that these objectives target distinct learning aspects, hence demonstrating strong complementarity.
>
> These details can be found in `Appendix F` of the revised manuscript.
>
> **W3. Systematic comparison of each component and source of gains.** To provide a consistent and directly comparable analysis, we re-ran the experiments corresponding to Figures 4(b), 5, and 6(a) under a unified setting with a *fixed random seed*, and provide a systematic ablation study in `Appendix G` of the revised manuscript as below.
>
> - baseline: 82.794
> - \+ P1 (Encoder-style, Naive): 83.115
> - \+ P2 (Feature-level Noise, Blk 2): 83.287
> - \+ P3 (Disruption): 83.410
>
> Together, the comparison demonstrates that each principle contributes meaningfully and that P3 is critical for realizing the overall gains.
>
> **W4. Which layer to finally apply noise to is not very clear.** As shown in Figure 5 of the original submission, injecting noise at Block 2 consistently yields optimal or near-optimal performance across all benchmarks, so we selected Block 2 as the layer for applying noise. To further highlight this, we boldface it in `Figure 5` of the revised manuscript.
>
> **Q1. Implementation details unclear. Other Contrastive MIM-based methods.**
> We provide expanded implementation details and full hyperparameter settings in `Appendix N` of the revised manuscript, and we note that the complete code necessary to reproduce experiments is already included in the Supplementary material.
>
> Regarding Contrastive MIM: Please see W2 (iBot [4], ConMIM [5], CAN [2]) and `Appendix F`.
>
> We also sincerely thank the reviewer for the references provided [1, 2]; they were helpful in contextualizing our work. Both references have been added.
>
>
> ----
> **References**
>
> [1] Multimodal Autoregressive Pre-training of Large Vision Encoders.
>
> [2] A SIMPLE, EFFICIENT AND SCALABLE CONTRASTIVE MASKED AUTOENCODER FOR LEARNING VISUAL REPRESENTATIONS
>
> [3] Learning transferable visual models from natural language supervision.
>
> [4] ibot: Image bert pre-training with online tokenizer.
>
> [5] Masked image modeling with denoising contrast.

---

> > ### Author Response · Authors · 2025-11-27
> > **Gentle Reminder for Reviewer vrdz**
> >
> > Thank you again for your thoughtful feedback on our submission. We would like to draw your attention to our responses to your concerns, along with the updated manuscript, as noted above.
> >
> > We would greatly appreciate it if you could take a moment to let us know of any further feedback or unresolved issues at your earliest convenience. *Your thoughtful comments have been instrumental in improving our work*, and we want to ensure that we address any remaining points before the deadline.
> >
> > We look forward to your continued engagement and appreciate your time and effort.
> >
> > Thank you.

---

### Author Response · Authors · 2025-11-19
**Paper Revision Summary**

We sincerely thank the Area Chair and all reviewers for their time and constructive feedback. We are encouraged that the reviewers found our empirical study thorough (vrdz, bBkf), our design choices carefully motivated by clear hypotheses (bBkf), the taxonomy and principles clear and actionable (vrdz, jK7h), our manuscript well-written and easy to follow (bBkf, jk7h), and the improvements substantial and consistent across diverse benchmarks (vrdz, bBkf, n4jd).

We would like to emphasize that **we have conducted all requested experiments and clarifications**, and have added these results in the Appendix of the revised manuscript. We note, however, that **all** of our new experiments are **in-line** with our findings, rooted from our initial detailed analysis in the original submission. We believe the new results strengthen our case, but the main substance of the paper, even revised, is the same as before, centered on the key insights we originally presented.

Key revisions (all reflected in the Appendix) include:

- **Appendix E, F, H**: Demonstration of generality and applicability; Extensions to Multimodal Pretraining (Appendix E), integration with Contrastive Learning (Appendix F), and complementarity with recent SOTA MIM methods (Appendix H).

- **Appendix M**: Comprehensive scalability analysis with larger models and longer schedules.

- **Appendix G, J**: Detailed analysis and efficiency; A systematic cumulative ablation study of each principle (Appendix G) and computational cost analysis (Appendix J).

- **Appendix I, K**: Additional visualizations: Layer-wise attention maps illustrating learned features (Appendix I) and affinity matrix visualizations demonstrating the effect of the disruption loss (Appendix K).

- **Appendix L, N**: Further clarifications on the relationship between MIM and Diffusion Models in our framework (Appendix L) and expanded implementation details/hyperparameter settings (Appendix N).

We believe these additions thoroughly address all reviewer concerns and significantly strengthen our manuscript. We thank the reviewers again for their engagement in improving our work.

---

### Author Response · Authors · 2025-12-03
**Summary of Submission #16244 for the Area Chair**

We first thank the area chair for the time and effort in handling our submission. Acknowledging the increased load under the revised ICLR workflow, and in the hope of reducing the overall evaluation burden, we provide the following concise summary of the paper’s novelty, the initial reviews, our comprehensive rebuttal, and the subsequent discussion.


# 1. Core Contribution and Novelty

This paper provides a systematic analysis of the Corruption-to-Reconstruction (C2R) paradigm in SSL. We investigate why prior attempts to combine masking and denoising (e.g., DiffMAE, MaskDiT) failed to improve recognition performance.

- **Novelty**: We introduce a clear taxonomy (Encoder/Decoder-style; Substitutive/Conjunctive) and derive *three synergistic principles* for effective C2R pretraining. The key insight is identifying and resolving the interference between masking and denoising objectives via explicit disentanglement (P3), which is crucial for unlocking significant performance gains (up to 8.1% over MIM baselines).

- **Impact**: Our principles are general, demonstrably improving vision-only MIM, SOTA MIM methods, and Multimodal frameworks (CLIP, AIM-v2).

# 2. Review Summary and Positive Feedback

Initial Scores: **6, 6, 4, 4** (Confidence: **5, 5, 3, 3**). Reviewers universally praised the paper's systematic approach, clarity, and empirical rigor.

- **R-vrdz (6)**: Highlighted the "thorough empirical study," "clear taxonomy," "actionable principles," and "substantial improvements."
- **R-bBkf (6)**: Noted the "carefully motivated" design choices, "consistent improvements," and that the paper is "well-written and structured."
- **R-jK7h (4)**: Appreciated the "clear definition," "controlled comparisons" supporting causal conclusions, and "clean design principles."
- **R-n4jd (4)**: Found the problem "interesting and important" and the study "systematic."

# 3. Addressing Key Concerns and Rebuttal Highlights

We addressed all concerns raised by the reviewers through extensive experimentation and clarification, added to the revised Appendix.

1) **[W] Component-wise Validation (vrdz, jk7h) & Motivation for P1 (jK7h):**
- *Action*: Provided a cumulative ablation (Appx. G) demonstrating that the principles are synergistic. We reinforced P1's motivation with empirical evidence (Appx. A) showing P2/P3 are ineffective in Decoder-style architectures.

2) **[W] Impact / Multimodal (n4jd, vrdz):**
- *Action*: Demonstrated the generality of our principles with experiments on Multimodal frameworks (AIM-v2, CLIP), showing clear improvements (Appx. E).

3) **[W] Scalability (jK7h):** (R-jK7h requested experiments beyond ViT-B/400ep.)
- *Action*: Provided extensive new experiments using ViT-L and longer schedules (800/1600 epochs), confirming consistent gains at scale (Appx. M).

4) **[W] Missing Comparisons (bBkf, vrdz):**
- *Action*: Compared with recent MIM (ColorMAE, HPM; Appx. H) and Contrastive methods (CAN, ConMIM; Appx. F), demonstrating complementarity and SOTA performance.

5) **[W] Visualizations/Analysis (bBkf, jK7h):**
- *Action*: Added feature visualizations (Appx. I), affinity matrix analysis showing interference mitigation (Appx. K), and computational cost analysis (Appx. J).

# 4. Positive Rebuttal Outcomes

Our comprehensive rebuttal successfully addressed the major concerns, leading to positive feedback.

- **R-jK7h (Initial Score 4):** Explicitly stated: *“I appreciate the authors’ efforts to address most of my comments with additional experiments and intuition, which substantially clarifies the method. I am therefore willing to raise my score.”*


Overall, we believe that, thanks to the reviewers’ constructive feedback, the revised manuscript is substantially stronger and now reasonably meets the bar for acceptance at ICLR 2026, and we hope it can be viewed favorably in the final decision.

---

### Meta-Review · Area_Chair_AYvZ · 2026-01-07

**Summary:**

This paper presents a corruption-based approach to improve the performance of self-supervised visual pre-training, especially the masked image modeling pipeline.

While the paper shows competitive performance on ImageNet-1K and other downstream datasets, the reviewers were mainly concerned about the following weaknesses.
* The paper is studying a relatively old topic. With the emergence of LLMs and multimodal LLMs, studying visual-only pre-training is no longer of sufficient interest to the community.
* The proposed algorithm contributes more to engineering -- the main algorithm and conclusions are mostly known.
* The results were not compared against recent methods.

**Reviewer Concerns:**

The authors addressed the above concerns. They provided new comparisons to show the proposed algorithm is competitive. However, the first two concerns cannot be well addressed. The AC tends to agree with the first concern -- the timeliness of the research is a major concern.

**Reviewer Scores:**

The initial score is 4/4/6/6, a borderline case. It is a bit difficult to guess how the reviewers would respond to the rebuttal if they had chance. On the one hand, the qualitative performance is promising and the authors added a comparison to recent methods. On the other hand, the AC agrees that the topic is relatively old and the overall technical contribution is incremental. Based on the above, the AC assumes that all reviewers will keep the original scores unchanged, and the AC tends to reject the paper because of its limited impact on the community.

---

### Decision · Program_Chairs · 2026-01-26

Reject